# Effects of Bacterial CLPB Protein Fragments on Food Intake and PYY Secretion

**DOI:** 10.3390/nu13072223

**Published:** 2021-06-29

**Authors:** Manon Dominique, Nicolas Lucas, Romain Legrand, Illona-Marie Bouleté, Christine Bôle-Feysot, Camille Deroissart, Fatima Léon, Séverine Nobis, Jean-Claude do Rego, Grégory Lambert, Pierre Déchelotte

**Affiliations:** 1TargEDys SA, 76183 Rouen, France; manon.dominique2@hotmail.com (M.D.); lucasnicolas@hotmail.fr (N.L.); legrandromain@hotmail.fr (R.L.); illonaboulete@yahoo.fr (I.-M.B.); camillederoissart@gmail.com (C.D.); glambert@targedys.com (G.L.); 2Inserm UMR1073, Nutrition, Gut and Brain Laboratory, University of Rouen Normandy, 76183 Rouen, France; christine.bole-feysot@univ-rouen.fr; 3Institute for Research and Innovation in Biomedicine (IRIB), University of Rouen Normandy, 76183 Rouen, France; fati.jpl@free.fr (F.L.); severine-nobis@orange.fr (S.N.); jean-claude.dorego@univ-rouen.fr (J.-C.d.R.); 4Animal Behavior Platform, Service Commun d’Analyse Comportementale (SCAC), University of Rouen Normandy, 76183 Rouen, France; 5Rouen University Hospital, CHU Charles Nicolle, 76183 Rouen, France

**Keywords:** caseinolytic peptidase B, fragments, food intake, peptide YY, in vivo

## Abstract

CLPB (Caseinolytic peptidase B) protein is a conformational mimetic of α-MSH, an anorectic hormone. Previous in vivo studies have already shown the potential effect of CLPB protein on food intake and on the production of peptide YY (PYY) by injection of *E. coli* wild type (WT) or *E. coli* ΔClpB. However, until now, no study has shown its direct effect on food intake. Furthermore, this protein can fragment naturally. Therefore, the aim of this study was (i) to evaluate the in vitro effects of CLPB fragments on PYY production; and (ii) to test the in vivo effects of a CLPB fragment sharing molecular mimicry with α-MSH (CLPB25) compared to natural fragments of the CLPB protein (CLPB96). To do that, a primary culture of intestinal mucosal cells from male Sprague–Dawley rats was incubated with proteins extracted from *E. coli* WT and ΔCLPB after fragmentation with trypsin or after a heat treatment of the CLPB protein. PYY secretion was measured by ELISA. CLPB fragments were analyzed by Western Blot using anti-α-MSH antibodies. In vivo effects of the CLPB protein on food intake were evaluated by intraperitoneal injections in male C57Bl/6 and ob/ob mice using the BioDAQ^®^ system. The natural CLPB96 fragmentation increased PYY production in vitro and significantly decreased cumulative food intake from 2 h in C57Bl/6 and ob/ob mice on the contrary to CLPB25. Therefore, the anorexigenic effect of CLPB is likely the consequence of enhanced PYY secretion.

## 1. Introduction

Obesity is a major public health problem. In 2016, more than 650 million adults (>18 years old) were considered obese [1]. According to the World Health Organization (WHO, Geneva, Switzerland), obesity is defined by a BMI ≥ 30 kg/m^2^, significantly affecting the quality of life of patients [2] with a risk factor for many diseases, such as type II diabetes [3], cancers [4], and cardiovascular diseases [5]. The composition of the microbiota is recognized to play a role in the pathophysiology of obesity. The colonization of germ-free mice with a gut microbiota harvested from obese mice resulted in an obese phenotype [6]. Other studies also highlighted that obese patients or ob/ob mice (a genetic model of obesity) exhibit a dysbiosis with a decrease of the Bacteroidetes/Firmicutes ratio [7,8] and a decrease of the archaea *Methanobrevibacter smithii* and of the bacteria *Escherichia coli* (*E. coli*) [9]. Consequently, under pathological conditions, the body adapts and reacts to the various associated disturbances, modifying the diversity, the composition, and the molecules secreted by the intestinal microbiota [10]. Previous studies from our group [11] have highlighted that *E. coli*, bacteria from the Enterobacteriaceae family, produces the caseinolytic peptidase B (CLPB) protein, which shares a common epitope of six amino acids with α-melanocyte-stimulating hormone (α-MSH), an anorexigenic neuropeptide [12,13]. This neuropeptide has central and peripheral effects: α-MSH present in the bloodstream can activate the melanocortin 4 receptor (MC4R) present (i) on anorexigenic proopiomelanocortin (POMC) neuronal populations [12] or (ii) on intestinal enteroendocrine L cells [14] to secrete Peptide YY (PYY). This confirms active communication along the “microbiota—intestine—brain” axis [15,16]. Accordingly, we previously suggested anorexigenic reactions to CLPB protein, similar to that with α-MSH in *E. coli* K12 WT (producer of CLPB protein) while administration of *E. coli* depleted in CLPB protein (*E. coli* ΔCLPB), not producing the CLPB protein, did not influence food intake or body composition [11,17]. In addition, intraperitoneal injections of proteins from *E. coli* WT in rats decreased food intake and induced an increase of the PYY level in the plasma [18] suggesting that the effect of CLPB protein could be at least in part mediated by PYY secretion. A recent in vitro study provided the first proof of a dose-dependent secretion of PYY by cultured rat intestinal cells treated with the CLPB protein [19]. In a pilot clinical study, the CLPB protein was also found at an increased level in the plasma of patients with eating disorders [20]. This increase of the CLPB protein level was also observed in a mouse model of anorexia [21]. Through its molecular mimicry with α-MSH, the plasma CLPB protein may modulate the mechanisms of appetite directly or by modulating the action of anti-α-MSH and anti-CLPB antibodies [11,22]. Another bacterium from the *Enterobacteriales* family, *Hafnia alvei* (*H. alvei*), also produces the CLPB protein and exerts anorexigenic effects in preclinical models of obesity [17]. Thus, all these studies suggest an inhibitory effect of the isolated CLPB protein on food intake, which, however, has not been directly demonstrated so far. In addition, Mock and his collaborators showed that the CLPB protein also has the capacity to fragment naturally [23]. However, it is not known whether the CLPB fragments generated by the microbiota can also stimulate the release of PYY and glucagon-like peptide-1 (GLP-1) and have a direct effect on food intake. To test this hypothesis, we investigated whether fragments of the CLPB protein generated by thermal or enzymatic shock could stimulate the in vitro production of PYY by cultured rat intestinal cells. These in vitro results suggest an anorexigenic effect of these CLPB fragments on food intake. That is why, in order to search for the fragments of CLPB responsible for this effect, analyses by Western blot were carried out, thus revealing the presence of a fragment of CLPB sharing a molecular mimicry with α-MSH (CLPB25). Then, the effects on the food intake of this fragment were evaluated in vivo in mice and they were compared with the in vivo effects of a naturally fragmented CLPB protein (CLPB96). All of these experiences have enabled us to demonstrate, for the first time, a direct in vivo anorexigenic effect of natural fragments of the CLPB96 but not of the CLPB25 fragment, as we initially thought. In addition, our in vitro studies suggest that this effect could be linked by a production of PYY.

## 2. Materials and Methods

### 2.1. Animals and Diets

Seven-week-old male Sprague–Dawley rats and 7-week-old male C57Bl/6 and ob/ob mice were purchased from Janvier Labs (Genest-Saint-Isle, France) and were kept for 1 week in rat or mice standard holding cages, in a specialized animal facility, to acclimate them to the environmental conditions: 22 ± 3 °C, relative humidity 40 ± 20%, and a 12 h light/dark cycle. Rats and mice were housed by two and three per cage, respectively. All of the experimental procedures were approved by the Local Ethical Committee of Normandy (approval no. 6701-2016083016464059 v5). All animals (mice and rats) were given ad libitum access to drinking water and food (Standard Diet, Kliba Nafag, Germany, ID: 3430). The commercial diet is composed of the following major nutrients: 880 g/kg of dry matter, 185 g/kg of protein, 45 g/kg of fat and fiber, 63 g/kg of ash, 542 g/kg of nitrogen-free extract (NFE), 16.1 MJ/kg of gross energy, 13.2 MJ/kg of convertible energy, and 35 g/kg of starch.

### 2.2. Bacterial Culture

*E. coli* WT and *E. coli* ΔCLPB (first generated by the Bernd Bukau’s laboratory (Center for Molecular Biology, Heidelberg University, Germany), then cultivated in the laboratory) were cultivated at 37 °C in Mueller–Hinton (MH) medium (Sigma-Aldrich, St. Louis, MO, USA). The growth rate was monitored every hour by measuring the optical density at 600 nm using a spectrophotometer (BioMate, ThermoElectron Corporation, Waltham, MA, USA) until reaching of the stationary phase (7 h). After a centrifugation step (2254× *g*, 30 min, 4 °C), the bacterial pellet was used for protein extraction. 

### 2.3. Bacterial Proteins Extraction

Bacterial pellet was dissolved in an extraction buffer (PBS + 1% of a protease inhibitor cocktail, Sigma, Poole, UK), sonicated (20% amplitude for 30 s), and centrifuged (12,000× *g*, 5 min, 4 °C) to separate cell debris from the cytoplasmic content. The supernatant was collected and stored at −80 °C if the analysis was not immediate.

### 2.4. Proteins Fragmentation

Proteins from *E. coli* WT and *E. coli* ΔCLPB (15 ng/µL) were fragmented with 0.25% trypsin without EDTA (CE. 3.4.21.4, Sigma) for 20 min at 37 °C (enzymatic fragmentation). Recombinant CLPB protein (CLPB96) purified by the chromatography method and at a concentration of 10.6 mg/mL was fragmented by a heat shock (30 min at 45 °C in a water bath) (thermal fragmentation). After incubation, the fragmentation was immediately stopped by placing tubes on ice. No treatment was applied to the CLPB96 protein, in order to analyze its natural fragmentation.

### 2.5. CLPB96 and CLPB25 Production

CLPB96 was produced by Delphi Genetics (Charleroi, Belgium, UniProtKB-P63284 CLPB_ECOLI). CLPB25 was produced by Delphi Genetics based on previously published data of its location on the CLPB96 sequence (536–756 aa) [24]. CLPB96 and CLPB25 were purified by chromatography method at a final concentration of 0.96 mg/mL and 0.28 mg/mL respectively.

### 2.6. CLPB Fragments Identification by Western Blot

Immunoblots were performed with the heat or enzymatic treated CLPB96 proteins. After fragmentation, CLPB96 was separated on a 20% polyacrylamide SDS-PAGE gel in a Tris-Glycine buffer (Biorad, Hercules, CA, USA) without the addition of β-mercaptoethanol. After separation, proteins were transferred onto a nitrocellulose membrane (GE Healthcare, Orsay, France), which was blocked for 1 h at room temperature with 5% (*w*/*v*) BSA (Bovine Serum Albumine) in TBST (10 mmol/L, pH 8, 150 mmol/L NaCl) + 0.05% (*w*/*v*) Tween 20. Then, the membrane was incubated overnight at 4 °C with anti-α-MSH antibodies (1:1000, Phoenix Pharmaceuticals Inc., Burlingame, CA, USA). The membrane was then washed three times and the peroxidase reaction was revealed using the ECL detection kit (GE Healthcare). After revelation, protein bands were compared to a molecular weight standard (Precision Plus Protein^TM^ Standards, Biorad) and films were scanned using ImageScanner III (GE Healthcare). To analyze the natural fragmentation and reveal all the fragments of the CLPB protein, CLPB96 was separated in another immunoblot without the addition of β-mercaptoethanol, as previously explained. After migration, proteins and fragments were stained with Coomassie Brilliant Blue R-250 staining solution (Biorad) according to the manufacturer’s instructions. 

Several 3D visualizations of the CLPB protein and fragments were carried out on PyMOL Software from the crystallization model of the CLPB protein (PDB: 4D2U) published on the RCSB website (https://www.rcsb.org (accessed on 25 June 2021)).

### 2.7. Rat Intestinal Cells Primary Culture and Gut Hormone Secretion

The protocol of intestinal cells primary culture was adapted from “Colonic culture preparation” from Psichas et al. [25]. Rats were killed by decapitation, and the colon and ileum were taken and washed firstly with an ice-cold PBS and secondly with ice-cold Leibowitz L-15 medium (Sigma-Aldrich). The intestinal tissues were then scraped, and the scraping digested with 0.4 mg/mL collagenase XI (Sigma) in high-glucose DMEM (1% L-Glutamine, 1% Penicillin, 1% Streptomycin, and 1% of non-essential amino acids) for 10 min at 37 °C. Cells suspensions were centrifuged (10 min, 400× *g*) and the pellets were resuspended in High-Glucose DMEM (the same as previously but with 10% fetal bovine serum (FBS). Cell suspensions were finally filtered through a nylon mesh (pore size 100 µm, Merck Millipore, Burlington, MS, USA) and put onto 24-well, 1% Matrigel-Coated Plates (Corning, NY, USA). The plates were incubated overnight at 37 °C in an atmosphere of 95% O_2_ and 5% CO_2_.

After 24 h of culture, intestinal cells were placed in a water bath (for experimental ease). The culture medium was gently removed, and a secretion buffer was added to the plates (4 mM KCl, 138 mM NaCl, 1.2 mM NaHCO_3_, 1.2 mM NaH_2_PO_4_, 2.6 mM CaCl_2_, 1.2 mM MgCl_2_, and 10 mM HEPES adjusted to pH 7.4 with NaOH), which was incubated for 20 min. A volume of 200 µL of total protein extracts from *E. coli* WT or *E. coli* ΔCLPB at 15 ng/µL and fragmented by trypsin (20 min, 37 °C) were included in the secretion buffer of each group (*n* = 4). Additionally, a volume of 200 µL of CLPB96 (Delphi Genetics) at two concentrations: 12.5 nM and 125 nM, fragmented by thermal shock (30 min at 45 °C) was added to the secretion buffer (*n* = 7). Cells incubated with PBS were used as control. After 20 min of incubation, the buffer was removed and cells were treated with a lysis buffer (50 mMol/L Tris-HCl, 150 mMol/L NaCl, 1% IGEPAL CA-630, 0.5% deoxycholic acid + protease inhibitor cocktail (Sigma)). Cells were then collected with a cell scraper and the lysates were centrifuged (12,000× *g*, 20 min) and stored at −80 °C until the PYY assay using a PYY (3–36) (Rat, Mouse, Porcine, Canine)—Fluorescent EIA Kit (sensibility range of 0–10,000 pg/mL) (Phoenix Pharmaceuticals Inc.) according to the manufacturer’s instructions.

### 2.8. CLPB25 Purification

CLPB25 purification was confirmed by Western Blot. CLPB25 was analyzed on a 20% polyacrylamide SDS-Page gel in a Tris-Glycine buffer (Biorad) in non-denaturant (without β-mercaptoethanol) or denaturant (with β-mercaptoethanol) conditions. After migration, proteins were transferred onto a nitrocellulose membrane (GE Healthcare) and the membrane was washed twice with distilled water. Then, the membrane was labeled with Ponceau red labeling solution (30 mL distilled H_2_O, 0.3 mL of acetic acid, 33 mg of Ponceau red) for 5 min with gentle agitation. Then, the red labeling solution was removed and the membrane was washed with distilled water at least three times. After labeling, protein bands were compared to a molecular weight standard (Precision Plus, Biorad) and the membrane was scanned using ImageScanner III (GE Healthcare). This Western Blot revealed the presence of two fragments (at 50 kDa and 25 kDa) in a non-denaturant condition, with a higher intensity for the 50 kDa fragment (CLPB50) (Appendix A). This fragment is a dimer composed of two CLPB25 bound by a disulfide bridge. Therefore, the molecular weight of CLPB50 was used for the in vivo experiments.

### 2.9. Intraperitoneal Injections of CLPB96 and CLPB25 in Mice

C57Bl/6 and ob/ob mice received daily intraperitoneal injections for 11 days at 9:00 a.m (beginning of the dark phase) of either (1) physiological serum; (2) CLPB96 at 0.95 µg/mL; (3) CLPB25 at 0.50 µg/mL. The chosen concentrations correspond to a final dose of 2 pMol/injection. In vitro, this concentration has been shown to have a significant effect on PYY secretion [19]. Mice were placed in the individual BioDAQ mouse cages (Research Diets Inc., New Brunswick, NJ, Canada) to measure individual food intake. An acclimatation period was carried out to limit the stress of the mice associated this new environment. The mice were had a similar weight per group (Appendix A). Mice were kept on free water and food access. By this BioDaq system, the mice feeders are connected to a computer, which very accurately calculates the amount of food consumed by each mouse all the time over 24 h. The effects of the CLPB protein on food intake were evaluated 2 h after daily intraperitoneal injection and for 12 h. At the end of the experiment, mice were anaesthetized by ketamine/xylaxine intraperitoneal and euthanized by decapitation.

### 2.10. Statistical Analysis

All the hypotheses were specified before the data were collected. The analytic plan was pre-specified and any data-driven analyses are clearly identified and discussed appropriately. All data are shown as means ± standard error (SEM). All statistical calculations were performed using GraphPad Prism 6.0 Software (GraphPad Software Inc., San Diego, CA, USA). Normality was evaluated using the Kolmogorov–Smirnov test. One-way ANOVA was performed to analyze PYY secretion, and the differences between groups were analyzed by using Dunn’s post-test for multiple comparisons. Two-way ANOVA was used to analyze food intake after CLPB protein treatment, followed by Sidak’s post-test for multiple comparisons. To analyze the effect of cumulative food intake at 120 min after CLPB protein treatment, the unpaired t-test and Mann–Whitney test were used when appropriate to distribution. A *p*-value of *p* < 0.05 (represented by the * symbol) was considered statistically significant. A *p* value of *p* < 0.10 (represented by the # symbol) was considered a statistical trend.

## 3. Results

### 3.1. Bacteria and CLPB96 Effects on PYY Secretion

*E. coli* WT proteins (15 ng/µL) fragmented after incubation with trypsin significantly stimulated PYY secretion compared to control (*p* < 0.05) and with a tendency compared to *E. coli* ΔCLPB (*p* < 0.10) (Figure 1A). Heat-treated CLPB96 (45 °C for 30 min) at 12.5 nM and 125 nM increased (*p* < 0.10 and *p* < 0.05, respectively) PYY secretion (Figure 1B). No significant difference was observed between 12.5 nM and 125 nM CLPB protein concentrations.

### 3.2. CLPB25 Identification

The Western Blot of untreated CLPB96 protein, stained with Coomassie blue, revealed a band of strong intensity at 96 kDa but also several other fragments of different molecular weight (Figure 2A). The Western Blot of CLPB96 after natural, enzymatic (trypsin, 37 °C for 20 min), or thermal (45 °C for 30 min) fragmentation highlighted a CLPB fragment at a molecular weight of 25 kDa, revealed with polyclonal anti-α-MSH antibodies (Figure 2B).

### 3.3. Localization of CLPB25

The localization of CLPB25 was determined according to the previously published sequence by Mogk (24). The sequence alignment between the Mogk fragment and CLPB96 (UniProtKB-P63284 CLPB_ECOLI) showed that this fragment was located at the center of the CLPB protein and that it contained the sequence mimicry of α-MSH in its N-terminal part (Figure 2C). A 3D visualization of the CLPB protein in its hexameric form, carried out using PyMOL Software, showed that the CLPB25 monomers surround the central pore of the protein, lying in an active site for protein disaggregation [26] (Figure 3A–C). A 3D visualization of CLPB96 confirmed that the amino acids constituting this pharmacophore motif E—RW-G-PV were located in the center and in AAA-2 domains useful for ATP binding and hydrolysis and is accessible to the surrounding interactions (Figure 3D–F).

### 3.4. In Vivo Effects of CLPB96 and CLPB25 in C57Bl/6 Mice

Daily injections of CLPB96 at 0.95 µg/mL in C57Bl/6 mice began to significantly decrease cumulative food intake from 75 min after injection (*p* < 0.05) (Figure 4A). On average, cumulative food intake 2 h after each injection of CLPB96 over 11 days was reduced by 25% compared to control mice (*p* < 0.001) (Figure 4B). The effect of CLPB96 injections lasted approximately 4 h and was followed by a rebound compensatory effect until the start of the day phase (Figure 4C). Daily injections of CLPB25 at 0.50 µg/mL in C57Bl/6 mice did not decrease cumulative food intake during the analysis (2 h after injection) (Figure 4D,E).

### 3.5. In Vivo Effects of CLPB96 and CLPB25 in ob/ob Mice

Daily injections of CLPB96 at 0.95 µg/mL in ob/ob mice significantly decreased cumulative food intake from 120 min after the injection (*p* < 0.05) (Figure 5A). On average, cumulative food intake 2 h after each injection of CLPB96 over 11 days was reduced by 22% compared to control mice (*p* < 0.001) (Figure 5B) and lasted approximately 4.5 h (Figure 5C). Daily injections of CLPB25 at 0.50 µg/mL in ob/ob mice did not decrease cumulative food intake during the time of the analysis (2 h after injection) (Figure 5D,E).

## 4. Discussion

In this study, we demonstrated, for the first time, a direct in vivo anorexigenic effect of natural fragments of the CLPB96 protein in mice. We also highlighted an increase of PYY secretion in response to fragments of CLPB96 (generated by enzymatic or thermal fragmentation) by the enteroendocrine intestinal cells in an in vitro model. Finally, among these fragments, we identified a fragment of interest due to its molecular mimicry with α-MSH, which did not turn out to influence in vivo cumulative food intake. Previous studies reported that PYY secretion from rat intestinal enteroendocrine cells was dose dependent on the administered CLPB protein and that bacterial proteins of *E. coli* WT, containing the CLPB protein, also stimulated PYY secretion in this in vitro model while bacterial proteins of *E. coli* ΔCLPB did not affect PYY secretion [19]. Now, it is well established that enteroendocrine L-cells secrete satietogenic hormones, such as PYY or GLP-1 [27] and that PYY production may be elicited at least in part via activation of MC4R stimulated by α-MSH. For example, the application of α-MSH on the basolateral mucosa of mice depleted of the MC4R receptor (MC4R−/−) did not induce any secretion of PYY [14]. Indeed, previous studies have revealed direct intermolecular contacts between amino acids belonging to α-MSH and MC4R: W9/F261; R8/E100, D122, D126; F7/F184; and H6/Y268, respectively. These amino acids constitute the pharmacophore tetrapeptide motif “HFRW”, which is necessary for MC4R recognition and activation [28]. A study has shown that this pharmacophore motif is located between the extracellular ends of the transmembrane helices on MC4R [28]. In addition, a recent study has shown that Ca^2+^ stabilized α-MSH-binding and function, and increased affinity of α-MSH for MC4R by acting as an endogenous cofactor for the binding of α-MSH to MC4R [29]. Ca^2+^ binds to the MC4R receptor on the following amino acids: D122, E100, and D126 and binds with α-MSH using H-R-L residues. Therefore, with (1) amino acids common with α-MSH and essential for MC4R receptor activation and (2) a smaller 3D structure to avoid steric hindrance when binding to the receptor, we investigated the hypothesis that fragments from the CLPB protein could activate the MC4R receptor. At the moment, this mechanism has not yet been confirmed but structural information is not the only relevant element for receptor function [30]. In fact, MC4R activation requires the movement of the transmembrane helices relative to each other, leading to the emergence of the cavity between them on the cytoplasmic side of the receptor [31]. Precise crystallographic studies would be necessary in order to prove the binding of the ClpB protein to this MC4R receptor. In this study, to validate that the effects of the CLPB protein on PYY secretion could be mediated by MC4R, it would have been interesting to use an MC4R antagonist. Then, other studies using α-MSH as a control could have been interesting to compare its response to the secretion of PYY in the same in vitro model to that of the CLPB protein. In addition, in this study, Western Blot analysis of the CLPB protein after enzymatic fragmentation with trypsin or after heat-treatment revealed, with polyclonal anti-α-MSH antibodies, a high-intensity band of 25 kDa (CLPB25). This fragment could be the same as that detected by Mock after incubation of the CLPB protein with the subtilisin (EC 3.4.21.62) enzyme [24]. The identified fragment held the pharmacophore in the N-terminal position (Figure 2C and Figure 3E,F). Given this feature, we hypothesized that this fragment could be responsible for the anorexigenic effects of the CLPB protein. Due to its small size, this fragment could pass the intestinal barrier, reach the bloodstream, and act centrally [18]. Additionally, MC4R are also present at the basolateral level of intestinal enteroendocrine L cells [14]. Thus, according to our hypothesis, this CLPB fragment could also activate the intestinal MC4R, as previously explained, and induce the secretion of PYY, therefore contributing to the global anorexigenic effect. For this reason, we conducted in vivo studies on the effect of intraperitoneal injections of CLPB96 and CLPB25 fragments on cumulative food intake. Daily injections confirmed the anorexigenic effects of natural fragments of the CLPB96 protein with a 20% decrease of cumulative food intake. This effect lasted up to 4 h after injection, both in standard C57Bl/6 and in ob/ob mice. The less pronounced anorexigenic effect of natural fragments of CLPB96 in ob/ob mice could be related to the hyperphagic behavior of this model. Furthermore, cessation of the anorexigenic response to the CLPB protein after a few hours may reflect the degradation of CLPB protein and/or return to baseline of PYY secretion, as already described in the post-prandial phase [32]. To better understand the underlying mechanisms, pharmacokinetic studies would be needed to measure the half-life of this protein, using labeled CLPB96. Moreover, it would have been interesting to measure the concentration of PYY in the plasma of mice in basal state and 2 h after intraperitoneal injections of ClpB; however, this invasive approach would have disrupted their feeding behavior and, moreover, the secretion of PYY remains a transient postprandial response, which a single sample would not be able to detect.

However, we preferred not to carry out this measurement, because this invasive approach would have disrupted their feeding behavior.

In the present study, CLPB25 had no effect on cumulative food intake in mice. This could be explained by the nature of its sequence. In fact, the position of the pharmacophore of CLPB25 identified by sequence analysis showed that it is located in the N-terminal position of the CLPB protein, which can lead to its rapid degradation by exonucleases in intestinal medium. We supposed that the anorexigenic effect of the CLPB protein is likely to be related to the presence of the pharmacophore. This is consistent with in vivo results: The degradation of the pharmacophore can explain the lack of effect of CLPB25 on cumulative food intake compared to CLPB96 where the pharmacophore is located in the central position, which confers it protection against exonucleases. Indeed, even if the CLPB protein fragments naturally, it contains the 96 KDa fragment unlike CLPB25. In the future, additional in vivo experiment, with a modified formulation allowing protection of CLPB96 and of CLPB25, should be considered to determine if CLPB25 is really devoid of any anorexigenic effect and whether the anorexigenic effects of CLPB96 can be amplified. The limit of this article is the short duration of the anorexigenic effect after injection of CLPB96 in mice, followed by a compensatory effect [33], which can explain that no difference in body weight of standard C57Bl/6 or ob/ob mice was found at the end of the study (data not shown). Adapted timing of repeated dosage (e.g., every 4–6 h) or development of prolonged delivery formulations may allow a sustained anorexigenic effect. With this in mind, the continuous release of the CLPB protein into the intestinal lumen by bacteria such as *H. alvei*, given via the oral route, would already be an efficient way of achieving a significant effect on body weight [17]. However, further studies are needed to highlight the specific contribution of the CLPB protein to explain the anorexigenic effect observed with these bacteria.

Finally, it would be interesting to evaluate the in vivo anorexigenic effects of all the fragments of the CLPB protein, other than the CLPB25 one, to confirm our hypothesis that one of the fragments contains the anorexigenic effect of the CLPB protein. If a fragment is predominantly identified, its 3D conformation and its action on the MC4R receptor could be studied by crystallographic studies.

## 5. Conclusions

In this study, we confirmed an anorexigenic effect of the natural fragments of the CLPB protein on cumulative food intake. This brings additional arguments supporting the role of the gut microbiota and its derived proteins and metabolites in the regulation of appetite. Our data suggest that improved formulations of the CLPB protein and/or its fragments should be further evaluated for their potential utility in the therapeutic strategy against overweight and obesity.

## Figures and Tables

**Figure 1 nutrients-13-02223-f001:**
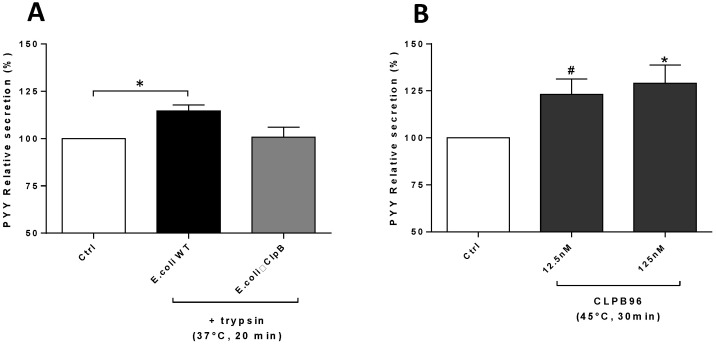
Effects of fragmentation of *E. coli* WT, *E. coli* ΔCLPB proteins, and CLPB96 on PYY secretion. PYY relative secretion (ng/µL) was measured (**A**) after incubation of total proteins of *E. coli* WT or *E. coli* ΔCLPB at 15 ng/mL, after enzymatic fragmentation by trypsin (37 °C, 20 min), *n* = 4, or (**B**) after incubation of CLPB96 at two concentrations: 12.5 nM and 125 nM, after thermal fragmentation (45 °C, 30 min), *n* = 7. Values are means ± standard error (SEM). (**A**,**B**) One-way ANOVA test with Dunn’s post-test for multiple comparisons. * *p* < 0.05; # *p* < 0.10 vs. control. PYY, Peptide YY; *E. coli*, *Escherichia coli*; CLPB96, recombinant CLPB protein: UniProtKB-P63284 CLPB_ECOLI.

**Figure 2 nutrients-13-02223-f002:**
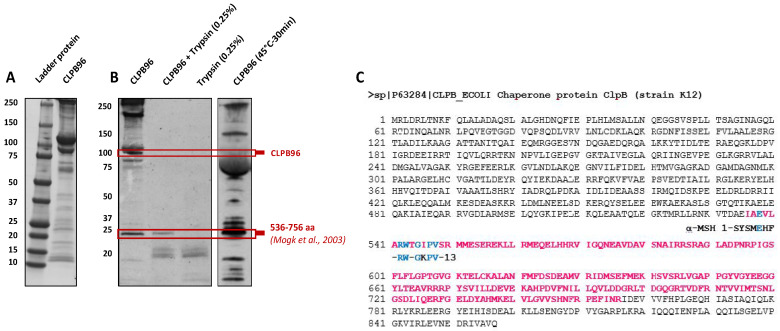
Natural fragmentation of the ClpB protein and identification of CLPB25 in the sequence of CLPB96. Identification by Western Blot of (**A**) natural fragments of CLPB96 stained with Coomassie blue and (**B**) of CLPB25 after enzymatic or thermal fragmentation revealed with polyclonal anti-α-MSH; (**C**) localization of CLPB25 in the sequence of CLPB96 (UniProtKB-P63284 CLPB_ECOLI) according to the previously published sequence [24].

**Figure 3 nutrients-13-02223-f003:**
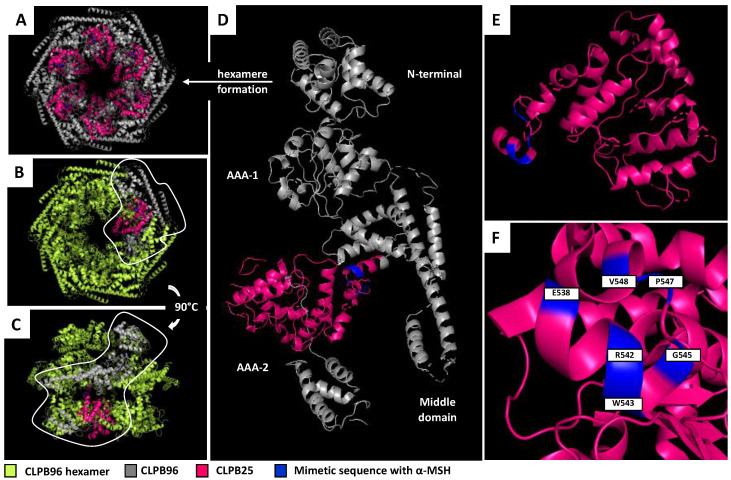
3D structures of CLPB96 and CLPB25 realized by PyMOL. The 3D visualizations of the CLPB protein were carried out on PyMOL Software from the crystallization model of the CLPB protein (PDB: 4D2U). 3D structure (**A**) of the CLPB protein in its hexameric form; (**B**) in top view; (**C**) in side view; (**D**) 3D structure of CLPB96; (**E**) of CLPB25 and (**F**) pharmacophore localization. CLPB96, recombinant CLPB protein: UniProtKB-P63284 CLPB_ECOLI; CLPB25, 25 kDa CLPB fragment: Location 536–756 aa on sequence UniProtKB-P63284 CLPB_ECOLI.

**Figure 4 nutrients-13-02223-f004:**
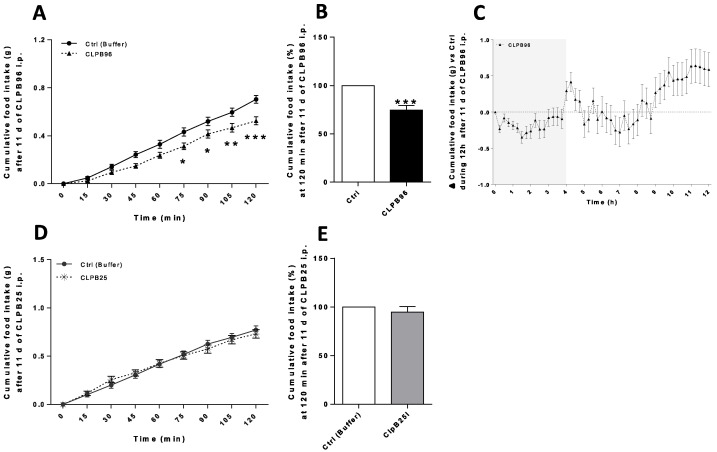
Effects of CLPB96 and CLPB25 on food intake in C57B1/6 2 h after daily intraperitoneal injection. Cumulative food intake (g) measured 2 h after daily injection (**A**) of CLPB96 or (**D**) CLPB25 in C57Bl/6 mice for 11 days at the end of experiment; (**B**,**E**) at 120 min and (**C**) Δ cumulative food intake for 12 h. Values are means ± standard error (SEM), *n* = 8; (**A**,**D**) 2-way ANOVA test with Sidak’s post-test; (**B**,**C**) Unpaired *t*-test. *** *p* < 0.001; ** *p* < 0.01; * *p* < 0.05 vs. Control. CLPB96, recombinant CLPB protein: UniProtKB-P63284 CLPB_ECOLI. CLPB25, 25 kDa CLPB fragment: Location 536–756 aa on sequence UniProtKB-P63284 CLPB_ECOLI.

**Figure 5 nutrients-13-02223-f005:**
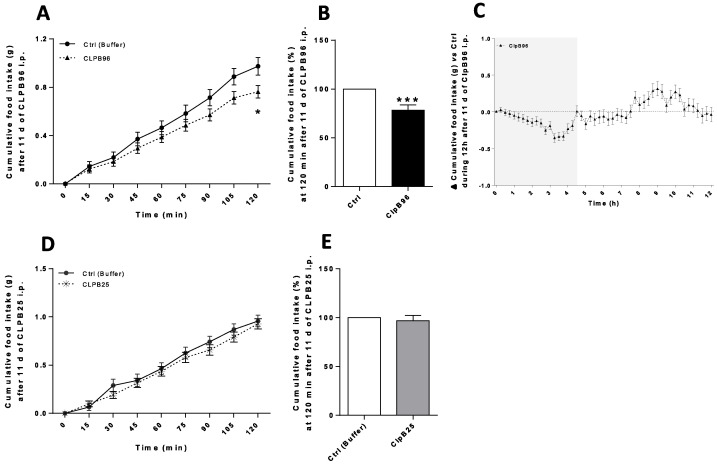
Effects of CLPB96 and CLPB25 on food intake in ob/ob mice 2 h after daily intraperitoneal injection. Cumulative food intake (g) measured 2 h after daily injection (**A**) of CLPB96 or (**D**) CLPB25 in C57Bl/6 mice for 11 days at the end of experiment; (**B**,**E**) at 120 min and (**C**) Δ cumulative food intake for 12 h. Values are means ± standard error (SEM), *n* = 8; (**A**,**D**) 2-way ANOVA test with Sidak’s post-test; (**B**) Mann and Whitney test; (**E**) Unpaired *t*-test. *** *p* < 0.001; * *p* < 0.05 vs. Control. CLPB96, recombinant CLPB protein: UniProtKB-P63284 CLPB_ECOLI. CLPB25, 25 kDa CLPB fragment: Location 536–756 aa on sequence UniProtKB-P63284 CLPB_ECOLI.

## Data Availability

Not applicable.

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
