# Peer review of "Effects of Bacterial CLPB Protein Fragments on Food Intake and PYY Secretion"

_nutrients, 2021, doi:10.3390/nu13072223_

Round 1
Reviewer 1 Report
[General comments]
The increase of obesity is a serious problem in many developed and developing countries. Since the close relationship between obesity and gut microbiota has been demonstrated, the improvement of gut microbiota is expected to control the body weight. The authors focused on the gut-brain axis and tried to use the gut bacteria-derived products for suppressing the appetite through increasing PYY in the intestine. This idea is interesting and the results in this research will be promising. However, there are several issues to investigate for the practical application of bacterial CLPB.
[Specific comments]
- The authors consider that CLPB produced by E. coli has the action similar to α-MSH in PYY induction through MC4R-triggering. In Fig.1, fragmented CLPB was subjected to the assay of PYY-inducing activity. To confirm the efficacy of fragmented CLPB in PYY-induction, comparison with α-MSF and undigested E. coli CLPB96 should be necessary.
- In Figs.4 and 5, the authors examined the in vivo effect of CLPB96 and CLPB25, and found that CLPB96, but not CLPB25, reduced the food intake of mice. As a hypothesis explaining no activity of CLPB25, the possibility of its easier degradation of CLPB25 by exonuclease is discussed. Why can exonuclease digest the CLPB25 peptide? If this is the case, do the authors think that CLPB25 still works as therapeutic way?
- Figs.4c and 5c give the interesting phenomena. Just after stopping CLPB96 injection, the food intake of mice rebound swiftly. This seems to show the safety of this peptide, but from the practical viewpoint the efficacy of CLPB96 may not be sufficient. How do the authors explain the transient reduction of appetite of mice triggered by CLPB96 injection?
- General concern is the antigenicity of bacterial protein. Is there no possibility that CLPB96 may induce the antibody which recognizes the endogenous α-MSF?
- As the authors discussed, they have already developed the promising product containing Hafnia alvei to treat obesity. What is the better point of E. coli-derived CLPB96 (injection agent) compared with H. alvei (oral supplement)?
Author Response
The increase of obesity is a serious problem in many developed and developing countries. Since the close relationship between obesity and gut microbiota has been demonstrated, the improvement of gut microbiota is expected to control the body weight. The authors focused on the gut-brain axis and tried to use the gut bacteria-derived products for suppressing the appetite through increasing PYY in the intestine. This idea is interesting and the results in this research will be promising. However, there are several issues to investigate for the practical application of bacterial CLPB.
- Answer: We agree that several issues must be investigated before bacterial ClpB may be used in practice; this is stated in the revised manuscript:
- Proving the binding of the ClpB protein to the MC4R receptor and their effects on the secretion of PYY;
- Measure the half-life of CLPB protein;
- Confirm the link between the ClpB protein and PYY secretion in vivo; The sentence “Moreover, it would be have been interesting to measure the concentration of PYY in the plasma of mice in basal state and 2h after intraperitoneal injections of ClpB[…] behavior” (Discussion);
- Repeat the same experiments in vivo but with a CLPB96 and CLP25 protein protected from degradation;
- Test repeated doses of CLPB protein in vivo;
- Analyze and compare anorectic effects of all the CLPB fragments.
The authors consider that CLPB produced by E. coli has the action similar to α-MSH in PYY induction through MC4R-triggering. In Fig.1, fragmented CLPB was subjected to the assay of PYY-inducing activity. To confirm the efficacy of fragmented CLPB in PYY-induction, comparison with α-MSH and undigested E. coli CLPB96 should be necessary.
- Answer: We agree with the comment of the reviewer. It would be interesting to use a-MSH as a control and to compare its in vitro effects on the PYY production. This could be done in future experiments and has been added in the discussion (in the beginning).
In Figs.4 and 5, the authors examined the in vivo effect of CLPB96 and CLPB25, and found that CLPB96, but not CLPB25, reduced the food intake of mice. As a hypothesis explaining no activity of CLPB25, the possibility of its easier degradation of CLPB25 by exonuclease is discussed. Why can exonuclease digest the CLPB25 peptide? If this is the case, do the authors think that CLPB25 still works as therapeutic way?
- Answer: we think that CLPB25 can be degraded by exonuclease because the CLPB protein is produced by intestinal bacteria in the gut, which is a very active enzymatic environment composed by bacteria and host enzymes.
Figs.4c and 5c give the interesting phenomena. Just after stopping CLPB96 injection, the food intake of mice rebound swiftly. This seems to show the safety of this peptide, but from the practical viewpoint the efficacy of CLPB96 may not be sufficient. How do the authors explain the transient reduction of appetite of mice triggered by CLPB96 injection?
- Answer: We believe that the anorectic effect of the CLPB96 protein may be generated in part by the mimetic sequence with a-MSH. If this is the case, CLPB96 unlike CLPB25 has its mimetic fragment in its central part which protects its anorectic action for a prolonged period. Once degraded, it will no longer have an effect, which may explain the transient reduction in appetite in mice triggered by the injection of CLPB96, with some kind of rebound effect after an acute administration. In contrast, in the vivo situation, if CLPB96 protein is released continuously (from bacteria or from controlled-release pharmaceutical formulations), the effect on food reduction is expected to be maintained, as shown in our rodents experiments (Legrand et al., 2020, Lucas et al., 2020).
General concern is the antigenicity of bacterial protein. Is there no possibility that CLPB96 may induce the antibody which recognizes the endogenous α-MSH?
- Answer: Yes, it’s possible, the antibodies can be induced by a-MSH and CLPB protein and recognize both proteins. Accordingly, our group previously hypothesized that anti-antibodies recognizing a-MSH may be involved in the mechanisms of eating disorders and obesity since a moderate increase of plasma levels of anti-ClpB IgG “cross-reactive” with a-MSH were observed in different groups of patients with eating disorders (Breton Int J ED). However, our most recent data (Galmiche et al Nutrients 2020) do not support different levels and affinity for these IgG between restrictive and compulsive patients. Thus, these antibodies seem to be indicative of the molecular mimicry but not to play a major role in the signaling of food intake by a-MSH.
As the authors discussed, they have already developed the promising product containing Hafnia alvei to treat obesity. What is the better point of E. coli-derived CLPB96 (injection agent) compared with H. alvei (oral supplement)?
- Answer: The better point of CLPB96 derived from coli (injection agent) compared to H. alvei (oral supplement) would be that IP (or sub-cutaneous in humans) E. coli-derived CLPB96 would reach directly the bloodstream and diffuse more quickly to the brain while H. alvei-derived CLPB96 may need to stimulate satietogenic hormones in the gut mucosa to exert its effects. Thus, injection of CLPB96 protein may be of interest to achieve a rapid effect to treat compulsive behaviours.
Reviewer 2 Report
This paper aims at testing the effect of CLBP protein in vivo on food intake in rodent model since this bacterial protein has been previously shown to stimulate PPY, a satietogenic peptide secreted by L cells in this distal gut. Different fragments are tested and particularly one (CLPB25) that presents mimicry with alpha-MSH, an anorexigenic neuropeptide.
Nevertheless, the main hypothesis and the strategy undertaken remain very unclear. The issues are not specified and we are not really convinced by the interest of testing different fragments of a CLPB protein which satiety effects are already well documented.
Major points are lacking to conclude that CLPB reduces food intake in standard and obese mice by increasing intestinal PPY production. Among them:
- concentration of PYY in plasma of mice in basal and 2h after i.p. of CLPB
- measurements of food intake during the postprandial state (for ex after the first nocturnal meal) since PYY is a satiety peptide which regulates food intake postprandially (inter-meal time, size if the second meal…). All these data of eating pattern are available in the BioDaq system.
Abstract
-Please provide a background and clearly define the purpose of the study.
-Is “injection” a pertinent keyword to define the paper?
-What is the sense of “CLPB96 fragmentation”: do the authors mean the natural fragmented state of this protein? Please clarify.
-PYY secretion results have been obtained in vitro on primary intestinal culture from rats whereas food intake has been recorded in mouse: please do not place these data as if they were obtained in the same model. In this sentence, “from 2h to 4h” in totally out of context, please precise or delete. Statistical analyses did not compare the data to CLPB25 but to the CTL group, please correct.
Introduction
What do the authors mean by a protein that fragments naturally? Is the CLBP protein still present in fragmented form in the lumen of the digestive tract? Do all known strains of E. coli in the intestinal microbiota produce CLBP? Under what conditions?
The sentence “Accordingly,….body composition” first suggests a reaction to CLPB and lasts with a experimental data. Please clarify.
The reference 19 is often cited in the manuscript but is unfortunately unpublished or incomplete in the reference list. Please update or precise as unpublished.
In order to understand what is at stake in this work, please explain clearly the difference between the naturally fragmented CLPB protein and the fragments produced by the microbiota (are they the same?). And what is the interest to study some fragments more than others? If the goal was to test only those produced by the microbiota (as mentioned), I don't understand the strategy using recombinant CLPB and heat shock.
As mentioned above, hypothesis and significance of the work must be stated.
The authors have to include the major result and the main conclusion of their study at the end of the introduction section.
Material and methods
Diet: Fiber is presented with fat: this is quite unusual and puzzling. Spell NFE.
Mice are not figured in the animal section and please specify their diet.
2.4 Source of total protein extract (mentioned in 2.7) should be placed here. This not clear if total protein extract was prepared from bacterial culture in the lab (2.2) or from Berns Bukau’s lab.
The objective of all these fragmentations procedure and source of protein used is totally unclear.
Why the authors used recombinant CLPB96 (fragmented or not?) and CLPB25 in vivo in mice whereas they used total protein extract (fragmented by trypsin) from E. coli and only recombinant CLPB96 (fragmented by thermal shock) in primary intestinal culture from rats?
2.7 Conditions of primary cultures must be more documented. I supposed culture was made at 37°C, so what is the point to put the cells in a water bath?
What are the volumes used for incubations? Are the cells polarized in these conditions of culture?
2.9 Please describe more accurately the BioDaq session. Time for mice to adapt, diet, inversed cycles…BioDaq is a continuous system to measure food intake so why the duration of registration was limited to 2h post i.p. and 12h after?
IP were performed at 9:00 am : were the animals in dark or diurnal phase? Food intake in higher in dark phase in rodents so it is important to point it out.
Reference for molecular weight in Western blots are lacking.
Results
3.1. Please homogenize statistical figure. The tendency in fig 1A does not clearly appears.
CTRL should have SEM since absolute values of PYY concentration in CTRL conditions are certainly different. Please could you indicate the sensibility of the ELISA test and the basal value of PPY concentration in the secretion buffer at the end of incubation (in CTRL)?
Figure 2. Please indicate where CLPB 96 stained on the blot.
3.4 In rodent, food intake is directly correlated to bodyweight. So, please, provide data of food intake in g/kg BW. Is Fig 4A representative of one day of registration (at the end of experiment) or the mean of cumulative FI during 2h post ip registered every day during 11 days? Please indicate in the legend or in the MM section.
Discussion
Authors should discuss the potential impact of their ip injection of CLPB protein on hypothalamic a-MSH neurons that regulate FI.
Postprandial effect of PYY should also be discussed in context of its satietogenic property. In view of previous studies published by their lab, authors should provide nuances on the regulation of short-term food intake by gastrointestinal peptide such as PYY of longer-term effects leading to eating disorders.
Reference list must be updated for references 17, 19 and 21 and corrected for 9.
Author Response
This paper aims at testing the effect of CLBP protein in vivo on food intake in rodent model since this bacterial protein has been previously shown to stimulate PPY, a satietogenic peptide secreted by L cells in this distal gut. Different fragments are tested and particularly one (CLPB25) that presents mimicry with alpha-MSH, an anorexigenic neuropeptide.
Nevertheless, the main hypothesis and the strategy undertaken remain very unclear. The issues are not specified and we are not really convinced by the interest of testing different fragments of a CLPB protein which satiety effects are already well documented.
- Answer: We have modified the manuscript according to this comment to clarify the interest of focusing on fragments in this new study.
Major points are lacking to conclude that CLPB reduces food intake in standard and obese mice by increasing intestinal PPY production. Among them:
- concentration of PYY in plasma of mice in basal and 2h after i.p. of CLPB
- measurements of food intake during the postprandial state (for ex after the first nocturnal meal) since PYY is a satiety peptide which regulates food intake postprandially (inter-meal time, size if the second meal…). All these data of eating pattern are available in the BioDaq system.
- Answer: Indeed, it would have been interesting to measure the concentration of PYY in the plasma of mice in basal state and 2h after i.p. of ClpB. However, this invasive approach would have disrupted their feeding behavior. Thus, we can only suggest that the observed effect of ClpB injection on reduction of food is related to PYY, based on our previous in vitro findings (Manon et al, Nutrients). This has been added in the discussion.
Abstract
-Please provide a background and clearly define the purpose of the study.
- Answer: For clarity, we have modified the abstract accordingly. We have added the sentence “an anorectic hormone. Previous in vivo studies have already shown the potential effect of CLPB protein on food intake and on the production of PYY by injection of coli WT or E. coliDClpB.”.
-Is “injection” a pertinent keyword to define the paper?
- Answer: This has been corrected, we have replaced the keyword “injection” by “in vivo” which indeed seems more appropriate.
-What is the sense of “CLPB96 fragmentation”: do the authors mean the natural fragmented state of this protein? Please clarify.
- Answer: This has been corrected. We have changed “CLPB96 fragmentation” by “natural CLPB96 fragmentation”.
-PYY secretion results have been obtained in vitro on primary intestinal culture from rats whereas food intake has been recorded in mouse: please do not place these data as if they were obtained in the same model. In this sentence, “from 2h to 4h” in totally out of context, please precise or delete. Statistical analyses did not compare the data to CLPB25 but to the CTL group, please correct.
- Answer: This has been corrected. We have modified the abstract by specifying that the results of PYY secretion were obtained with an in vitro
- For clarity, “from 2h to 4h” has been modified in the sentence. Indeed, statistical analyzes did not compare the data to Ctrl but to the CLPB25 fragment and revealed the effect of CLPB96 protein compared to CLPB25. So, we’ve changed the word “compared to” to “the contrary to” in the abstract.
Introduction
What do the authors mean by a protein that fragments naturally? Is the CLBP protein still present in fragmented form in the lumen of the digestive tract? Do all known strains of E. coli in the intestinal microbiota produce CLBP? Under what conditions?
- Answer: Yes, this protein fragments naturally without any parameters changed. But, at the moment, we don’t know what form or which fragments of CLPB are present in the lumen of the digestive tract. These are study perspectives on this project.
- Normally, all known strains of coli in the gut microbiota can produce the CLPB protein. However, because it is a heat shock protein, it appears to be produced in greater amounts during stress, but also in the presence of protein (like BSA) as we have shown in a previous article (Dominique et al., 2019).
The sentence “Accordingly,….body composition” first suggests a reaction to CLPB and lasts with a experimental data. Please clarify.
- Answer: Indeed, Tennoune et al., 2014 have shown that daily intragastric gavage of mice with coli WT, E. coliDClpB, α-MSH during 3 weeks were accompanied by a decrease in body weight and food intake in mice receiving E. coli WT. After this, food intake returned to control levels. This early study suggested an effect of the bacterial ClpB protein on the acute reduction of food intake. Another recent study (Dominique et al., 2019) also confirmed in vitro the potential of E. coli WT (containing the ClpB protein) on the secretion of PYY unlike E. coli WTDClpB.
The reference 19 is often cited in the manuscript but is unfortunately unpublished or incomplete in the reference list. Please update or precise as unpublished.
- Answer: The reference has been corrected.
In order to understand what is at stake in this work, please explain clearly the difference between the naturally fragmented CLPB protein and the fragments produced by the microbiota (are they the same?). And what is the interest to study some fragments more than others? If the goal was to test only those produced by the microbiota (as mentioned), I don't understand the strategy using recombinant CLPB and heat shock.
- Answer: With this study, we can’t say with certainty that the naturally fragmented CLPB fragments and the fragments produced by the microbiota are similar. But we hope that if and our study allows us to study all the possible fragmenting conditions that the CLPB96 protein could undergo: i) heat shock for possible temperature variations in the organism (especially because CLPB is a heat shock protein and that it can generate easier these fragments ii) in contact with trypsin, an enzyme that can be found in the intestine.
As mentioned above, hypothesis and significance of the work must be stated.
- Answer: Hypothesis and objectives of the work have been stated more clearly in the abstract.
The authors have to include the major result and the main conclusion of their study at the end of the introduction section.
- Answer: We have added the sentence “All of these experiences have enabled us to we demonstrated for the first time a direct in vivo anorexigenic effect of natural fragments of the CLPB96 and to confirm that this effect could be linked by a production of PYY” in the end of the Introduction section.
Material and methods
Diet: Fiber is presented with fat: this is quite unusual and puzzling. Spell NFE.
- Answer: NFE stands for nitrogen-free extract. This has been added in the manuscript.
Mice are not figured in the animal section and please specify their diet.
- Answer: Mice are listed in the animal section with the sentence « 7-week-old male Sprague-Dawley rats and 7-week-old male C57Bl/6 and ob/ob mice were purchased from Janvier Labs”, “rat or mice standard holding cages”. The diet is similar for mice or rat (section 2.1).
2.4 Source of total protein extract (mentioned in 2.7) should be placed here. This not clear if total protein extract was prepared from bacterial culture in the lab (2.2) or from Berns Bukau’s lab.
- Answer: This has been corrected. The source of total protein extract was generated in the first time by the Bernd Bukau’s laboratory (section 2.2) and after we have cultivated bacteria in the lab for our experiments. The following sentence has been added “first generated by the Bernd Bukau’s laboratory (Center for Molecular Biology, Heidelberg University, Germany), then cultivated in the laboratory) …” (section 2.2).
The objective of all these fragmentations procedure and source of protein used is totally unclear.
- Answer: We have used different type of fragmentation to study all types of fragmentation the CLPB protein could undergo (thermal fragmentation by its nature of heat shock protein and enzymatic fragmentation by trypsin as in the intestine). We have provided details in section 2.4.
Why the authors used recombinant CLPB96 (fragmented or not?) and CLPB25 in vivo in mice whereas they used total protein extract (fragmented by trypsin) from E. coli and only recombinant CLPB96 (fragmented by thermal shock) in primary intestinal culture from rats?
- Answer: We had already shown in a first study the effect of the recombinant CLPB protein (natural fragmentation) on PYY secretion in an in vitro model of rat primary intestinal cells (Dominique et al., 2019). These results had shown a dose effect between the concentration of the CLPB protein and the secretion of PYY produced by the intestinal cells. Therefore, we wanted to use this protein for in vivo studies to assess and confirm the link between this protein, PYY secretion and its anorectic effect in vivo.
- In the same previous study, we had already evaluated the effect of coli WT and E. coli DClpB proteins on PYY production. The objective of the present paper was to assess the effect of fragmentation on these same proteins and their impacts on PYY secretion.
- For CLPB25, it would have been interesting to test it in vitro on the intestinal cells of rats. However, we had first carried out the in vivo experiment and having had no effect on food intake. So, we didn’t wish to test its effect on rat intestinal cells.
2.7 Conditions of primary cultures must be more documented. I supposed culture was made at 37°C, so what is the point to put the cells in a water bath?
- Answer: The culture has been made at 37°C in a water bath, for experimental ease: to administrate CLPB protein and maintain the cells in best culture conditions.
The protocol of primary culture (section 2.7) has been more documented and we have added the paragraph to explain all the conditions used:” Rats were killed […] 5% CO2”.
What are the volumes used for incubations? Are the cells polarized in these conditions of culture?
- Answer: The incubations have been made in a volume of 200µl. We have added this information in section 2.7. Yes, we used culture plates coated with Matrigel which allows the cells to be polarized. In our study, the orientation of the cells is very important because the secretion of PYY takes place at the basal level.
2.9 Please describe more accurately the BioDaq session. Time for mice to adapt, diet, inversed cycles…BioDaq is a continuous system to measure food intake so why the duration of registration was limited to 2h post i.p. and 12h after?
- Answer: We decided to measure the food intake with the recording time limited to 2h post ip because it is the secretion time of the post-prandial peptide like PYY and in our study, we want to show the link between the anorectic effect of ClpB protein and PYY secretion.
- We also wanted to analyze the effect of the ClpB protein over 12 hours in order to assess the duration of its effect.
IP were performed at 9:00 am: were the animals in dark or diurnal phase? Food intake in higher in dark phase in rodents so it is important to point it out.
- Answer: IP were performed at 9:00 am, when the animals begin the diurnal phase. This was clarified in section 2.9.
Reference for molecular weight in Western blots are lacking.
- Answer: The molecular weights of the proteins has been defined using the Precision Plus ProteinTM Standards molecular marker (Biorad, Ca, USA). The reference is noted in section 2.6.
Results
3.1. Please homogenize statistical figure. The tendency in fig 1A does not clearly appears.
- Answer: For more homogenization of statistical results, the sentence "A P-value of P<0.05 (represented by the * symbol) was considered statistically significant. A P. value of P<0.10 (represented by the # symbol) was considered a statistical trend." has been modified and added in section 2.10.
CTRL should have SEM since absolute values of PYY concentration in CTRL conditions are certainly different. Please could you indicate the sensibility of the ELISA test and the basal value of PPY concentration in the secretion buffer at the end of incubation (in CTRL)?
- Answer: The values of PYY concentration of CTRL conditions remain quite similar, that’s why we have decided to present the results as a percentage of the control. The basal value of PYY concentration in the secretion buffer at the end of incubation (in CTRL) is in a range between 0.852 and 1.091. The values for each condition are respectively: 0.852; 1.091; 1.041; 1.016.
- The name and its sensitivity of the PYY Kit used has been precised in section 2.7.
Figure 2. Please indicate where CLPB 96 stained on the blot.
- Answer: CLPB96 stained has been indicated on the blot on figure 2.
3.4 In rodent, food intake is directly correlated to bodyweight. So, please, provide data of food intake in g/kg BW. Is Fig 4A representative of one day of registration (at the end of experiment) or the mean of cumulative FI during 2h post ip registered every day during 11 days? Please indicate in the legend or in the MM section.
- Answer: Upon receipt, the mice were given a one-week acclimatization period to eliminate any stress associated with transport. Following this acclimatization period, the groups of mice were defined according to their weight with the lowest SD. We agree that the food intake is directly correlated to bodyweight, but we believe that the homogeneity of the groups removes the bias that could be related to the different weights of the mice. In addition, the weights of mice were very similar.
Discussion
Authors should discuss the potential impact of their ip injection of CLPB protein on hypothalamic a-MSH neurons that regulate FI.
- Answer: Indeed, this impact has been discussed by the potential action of the ClpB protein on the MC4R receptor. If the anorectic action of the CLPB protein is carried by its mimetic sequence with α-MSH, we hypothesize that it could also activate this receptor. However, this hypothesis remains to be discussed because several other parameters must also be taken into account (stochimetric of the protein, presence of Ca2+).
Postprandial effect of PYY should also be discussed in context of its satietogenic property. In view of previous studies published by their lab, authors should provide nuances on the regulation of short-term food intake by gastrointestinal peptide such as PYY of longer-term effects leading to eating disorders.
- Answer: We agree with the comment of the reviewer. In this study, we wanted to focus in the first time on the short-term effect of PYY. But it would be interesting to study the effect of ClpB protein on the secretion of PYY of long-term.
Reference list must be updated for references 17, 19 and 21 and corrected for 9.
- Answer: All references have been updated.
Round 2
Reviewer 1 Report
Please check the following;
Page 2, the last sentence of Introduction: have enabled us to we demonstrated --- ⇒ have enabled us to demonstrate ---
Author Response
Page 2, the last sentence of Introduction: have enabled us to we demonstrated --- ⇒ have enabled us to demonstrate ---
- Answer: We have been changed the sentence in the Introduction.
Reviewer 2 Report
The authors responded to the minor points of my first report but the 3 major points remain to my opinion unresolved.
The authors did not respond to my first request for clarification of the main hypothesis and the strategy implemented to address it:
The hypothesis of the authors in the introduction section is : “it is not known whether the CLPB fragments generated by the microbiota can also stimulate the release of PYY and glucagon-like peptide-1 (GLP-1) and have a direct effect on food intake. To test this hypothesis, we investigated whether fragments of the CLPB protein generated by thermal or enzymatic shock could stimulate the in vitro production of PYY by cultured rat intestinal cells. Consequently, the effects on food intake of a CLPB fragment sharing molecular mimicry with α-MSH were evaluated in vivo in mice. These results were compared to the in vivo effects of a naturally fragmented CLPB protein.”
In the results section, trypsinized total protein from E.coli WT or DCLPB as well as thermally treated recombinant CLPB96 are shown to increase PYY in vitro. Since recombinant CLPB25 (the CLPB fragment sharing mimicry with α-MSH) is used in vivo for food intake measurements, its effect on PYY in vitro is expected. The answer of the authors to this point was “For CLPB25, it would have been interesting to test it in vitro on the intestinal cells of rats. However, we had first carried out the in vivo experiment and having had no effect on food intake. So, we didn’t wish to test its effect on rat intestinal cells.” This answer does not correspond to what is indicated in the introduction (underlined by the “consequently”).
Moreover, why didn’t the authors test the same CLPB preparations in vitro and in vivo? In vitro, trypsinized total E.coli proteins and heated recombinant CLPB96 are used. Western blots show the fragments generated by enzymatic and thermal treatments of recombinant CLPB. In vivo, untreated recombinant proteins are used.
Thus, the strategy set up to answer the authors' hypothesis remains for me much too confusing as it is.
A second major point was PYY in vivo data that are lacking. In their response, authors argue that “this invasive approach would have disrupted their feeding behavior”. I agree with this argument during the 11d of experimentation recording food intake during 12h post i.p. but animals were sacrificed at the end of the 11d-experiment when blood collection should have been done (2h after i.p. the next day to get at least one measurement of PYY concentration). This is why I think that the experimental design is not in line with the question asked. The authors themselves in their answer remind that their objective was " we wanted to use this protein for in vivo studies to assess and confirm the link between this protein, PYY secretion and its anorectic effect in vivo.”
A third point concerned Biodaq data. The effects of PYY on food intake must be completed by Biodaq data (e.g. time between 2 meals, size of the second meal...) which allow a detailed description of the prandial sequence regulated by PYY and not only the intake of food. The latter must, I insist, be related to the weight of animals. The fact that the authors "believe that the homogeneity of the groups removes the bias that could be related to the different weights of the mice" is not enough from my point of view. In the same way, it is not enough to say that the weights of the animals are “similar”, it must also be shown.
Author Response
The authors responded to the minor points of my first report but the 3 major points remain to my opinion unresolved.
The authors did not respond to my first request for clarification of the main hypothesis and the strategy implemented to address it:
The hypothesis of the authors in the introduction section is : “it is not known whether the CLPB fragments generated by the microbiota can also stimulate the release of PYY and glucagon-like peptide-1 (GLP-1) and have a direct effect on food intake. To test this hypothesis, we investigated whether fragments of the CLPB protein generated by thermal or enzymatic shock could stimulate the in vitro production of PYY by cultured rat intestinal cells. Consequently, the effects on food intake of a CLPB fragment sharing molecular mimicry with α-MSH were evaluated in vivo in mice. These results were compared to the in vivo effects of a naturally fragmented CLPB protein.”
In the results section, trypsinized total protein from E.coli WT or DCLPB as well as thermally treated recombinant CLPB96 are shown to increase PYY in vitro. Since recombinant CLPB25 (the CLPB fragment sharing mimicry with α-MSH) is used in vivo for food intake measurements, its effect on PYY in vitro is expected. The answer of the authors to this point was “For CLPB25, it would have been interesting to test it in vitro on the intestinal cells of rats. However, we had first carried out the in vivo experiment and having had no effect on food intake. So, we didn’t wish to test its effect on rat intestinal cells.” This answer does not correspond to what is indicated in the introduction (underlined by the “consequently”).
- Answer: We agree with the comment of the reviewer. All the end of the introduction has been changed consequently: “To test this hypothesis … by a production of PYY”.
Moreover, why didn’t the authors test the same CLPB preparations in vitro and in vivo? In vitro, trypsinized total E.coli proteins and heated recombinant CLPB96 are used. Western blots show the fragments generated by enzymatic and thermal treatments of recombinant CLPB. In vivo, untreated recombinant proteins are used.
Thus, the strategy set up to answer the authors' hypothesis remains for me much too confusing as it is.
- Answer: We agree with the comment of the reviewer. The in vitro studies aimed to assess the impact of different types of fragmentation that ClpB protein may undergo and to see their impact on the secretion of PYY. Western blot studies have confirmed the presence of a fragment of interest which, according to our initial hypothesis, could be the fragment responsible for the anorectic effect of the ClpB protein. The fact that several types of fragmentation (natural, thermal and enzymatic) were studied allowed us to verify the stability and the presence of the ClpB96 and ClpB25 forms. We decided to use ClpB96 (natural fragmentation) for our in vivo studies to evaluate the effect of our protein in "treatment" condition.
A second major point was PYY in vivo data that are lacking. In their response, authors argue that “this invasive approach would have disrupted their feeding behavior”. I agree with this argument during the 11d of experimentation recording food intake during 12h post i.p. but animals were sacrificed at the end of the 11d-experiment when blood collection should have been done (2h after i.p. the next day to get at least one measurement of PYY concentration). This is why I think that the experimental design is not in line with the question asked. The authors themselves in their answer remind that their objective was " we wanted to use this protein for in vivo studies to assess and confirm the link between this protein, PYY secretion and its anorectic effect in vivo.”
- Answer: The animals were sacrificed the 11d-experiment BUT 2h after i.p injection as indicated section 2.9 by “daily intraperitoneal injections for 11 days at 9:00a.m”. Thus, although we agree with the reviewer that it would have been interesting to measure the concentration of PYY in the plasma of mice after the CLPB injection, this would have interfered with the study design and disrupted food intake. In addition, PYY post-prandial response is a dynamic process and we doubt that a single sampling time would have detected a transient increase in PYY.
- In the discussion, we have been added the sentence “however this invasive approach would have disrupted their feeding behavior and, moreover, the secretion of PYY remains a transient postprandial response which a single sample wouldn’t not be able to detect”
A third point concerned Biodaq data. The effects of PYY on food intake must be completed by Biodaq data (e.g. time between 2 meals, size of the second meal...) c The latter must, I insist, be related to the weight of animals. The fact that the authors "believe that the homogeneity of the groups removes the bias that could be related to the different weights of the mice" is not enough from my point of view. In the same way, it is not enough to say that the weights of the animals are “similar”, it must also be shown.
- Answer: We tried in the time given to us to analyze the data from Biodaq (time between the first 2 meals, comparison of the size of the second meal compared to the first…) which unfortunately didn’t give interesting results. The Biodaq system being a very sensitive system, some meals were counted with too much weight (the mouse having moved the food platform or put its legs or its body on it). This measurement does not reflect the actual weight of their meal at time t. This bias was taken into account in the analysis of the food intake because the data set is very important. However when we compare 2 meals, the data set is very small and we cannot exclude this value, which proves that the mouse is eating at this time t. So, the weight value will be wrong. This is why, we do not think that it is interesting to give these graphs, even if we agree with the reviewer, it could have given an idea of effects of PYY post-prandial.
- To answer at the question of the reviewer, it would have been interesting to compare the weight of the mice before injection of CLPB96 or CLP25, so we added a supplementary figure 2 with new graphs proving the homogeneity in each group.